# Additive Pattern Databases for Decoupled Search

**Silvan Sievers,**[1] **Daniel Gnad,**[2] **Álvaro Torralba**[3]

[1] University of Basel, Switzerland
[2] Linköping University, Sweden
[3] Aalborg University, Denmark
silvan.sievers@unibas.ch, daniel.gnad@liu.se, alto@cs.aau.dk

## Abstract

Abstraction heuristics are the state of the art in optimal classical planning as heuristic search. Despite their success for explicit-state search, though, abstraction heuristics are not available for decoupled state-space search, an orthogonal reduction technique that can lead to exponential savings by decomposing planning tasks. In this paper, we show how to compute pattern database (PDB) heuristics for decoupled states. The main challenge lies in how to additively employ multiple patterns, which is crucial for strong search guidance of the heuristics. We show that in the general case, for arbitrary collections of PDBs, computing the heuristic for a decoupled state is exponential in the number of leaf components of decoupled search. We derive several variants of decoupled PDB heuristics that allow to additively combine PDBs avoiding this blow-up and evaluate them empirically.

## Introduction

Classical planning (Ghallab, Nau, and Traverso 2004) is the problem of finding a sequence of deterministic actions that lead from a given initial world state to a state satisfying a desired goal specification. A popular approach for optimally solving classical planning tasks is heuristic state-space search (Pearl 1984; Bonet and Geffner 2001), and A* search in particular. The most commonly used type of search space representation is *explicit search*, where each state is represented individually. Another successful alternative is *symbolic search* (e.g. Torralba et al. 2017), where search states are not represented individually, but as sets of states, using symbolic data structures. More recently, Gnad and Hoffmann (2018) introduced a third variant called *decoupled search*. It entails factoring the variables of a planning task into a center and several leaf factors, with the property that leaf factors are conditionally independent. This allows searching on the center only and keeping track of the reached leaf states and their cost, for a center path.

State-of-the-art heuristics for optimal planning for explicit A* search are based on abstractions and combined with cost partitioning (e.g. Seipp and Helmert 2018; Seipp, Keller, and Helmert 2020; Sievers and Helmert 2021). For decoupled search, heuristics in general have so far been computed using a compilation introduced by Gnad and Hoffmann (2018). The compilation modifies the planning task based on the decoupled state for which the heuristic is com-

puted. It is "perfect in the limit", i.e., computing the perfect (explicit state) heuristic on the compiled task is equivalent to computing the perfect heuristic for the decoupled state. While this approach could in principle also be used for abstraction heuristics, it is infeasible in practice because abstraction heuristics are precomputed, which is impossible given that the compilation depends on the decoupled state the heuristic should be evaluated on.

In this work, we therefore define an alternative way of using an existing (explicit state) heuristic in decoupled search, which does not rely on the compilation, by enumerating all explicit states represented by a decoupled state. We show that this way of using the heuristic is at least as good as via the compilation. Since this new approach possibly is prohibitively expensive, in the remainder of this work, we focus on computing projections, which are the fundamental abstractions underlying *pattern database (PDB)* heuristics (Culberson and Schaeffer 1998; Edelkamp 2001), for decoupled search. We show how to avoid enumerating the exponentially many explicit states represented by a decoupled state when computing the exact PDB heuristic values for that state. Furthermore, we show how to admissibly sum up heuristic values of PDBs which are *additive* in the sense of the canonical PDB heuristic (Haslum et al. 2007) or which have been cost-partitioned using *saturated cost partitioning* (Seipp and Helmert 2018). We prove that exact admissible combination is an **NP**-complete problem and present two approximations where we impose restrictions on the patterns used for the PDB heuristics. Our experimental study shows that PDBs can yield strong performance in decoupled search, surpassing the previous state-of-the art in optimal decoupled search, LM-cut (Helmert and Domshlak 2009), for some variants. We conclude with a discussion of directions for future work.

## Background

### Classical Planning

We consider the SAS$^+$ planning formalism (Bäckström and Nebel 1995), which is based on *finite-domain state variables*. Let $\mathcal{V}$ be a finite set of variables $v$, each with a finite domain $\mathcal{D}(v)$. A *partial state* $s$ is an assignment to a subset of the variables, written $vars(s) \subset \mathcal{V}$. We write $s[v]$ for the value of $v$ in $s$. For a subset $\mathcal{V}' \subseteq \mathcal{V}$, $s[\mathcal{V}']$ denotes the re-

striction of $s$ onto $\mathcal{V}'$, i.e., the assignment to $\mathcal{V}' \cap vars(s)$ by $s$. We also treat partial states $s$ as sets of *facts* $v \mapsto d$ for all $v \in vars(s)$ and $d \in \mathcal{D}(v)$. If $vars(s) = \mathcal{V}$, then $s$ is called a *state*.

A SAS$^+$ planning task is defined as $\Pi = \langle \mathcal{V}, \mathcal{A}, I, G \rangle$. $\mathcal{V}$ is a finite set of finite-domain state variables. $\mathcal{A}$ is a finite set of *actions* $a = \langle \mathsf{pre}(a), \mathsf{eff}(a), c(a) \rangle$, where $\mathsf{pre}(a)$ and $\mathsf{eff}(a)$ are partial states called *precondition* and *effect* of $a$, and $c(a) \in \mathbb{R}^{0+}$ is the *cost* of $a$. $I$ is the *initial state* and $G$ is a partial state called the *goal*.

Action $a$ is *applicable* in partial state $s$ if $\mathsf{pre}(a) \subseteq s$. Applying it leads to the successor state $s[\![a]\!]$, where $s[\![a]\!][v] = \mathsf{eff}(a)[v]$ for all $v \in vars(\mathsf{eff}(a))$ and $s[\![a]\!][v] = s[v]$ for all $v \notin vars(\mathsf{eff}(a))$. An *s-plan* is a sequence $\pi$ of actions applicable in $I$ such that the resulting state $s[\![\pi]\!]$ is a *goal state*, i.e., $G \subseteq s[\![\pi]\!]$. A plan for $\Pi$ is an *I*-plan. The cost of $\pi$, $c(\pi)$, is the summed-up action cost. Optimally solving a planning task means finding an *optimal* plan, i.e., one of minimal cost, or showing that no plan exists. We write $S(\Pi)$ for the set of states defined over $\mathcal{V}$ of $\Pi$.

## Pattern Database Heuristics

To solve planning tasks $\Pi$ optimally, we use the A$^*$ algorithm (Hart, Nilsson, and Raphael 1968) with an admissible heuristic. A *heuristic* $h$ maps a state $s$ of the task $\Pi$ to an estimate $h(s) \in \mathbb{R}^{0+} \cup \{\infty\}$ of the cost of reaching a goal. It is *admissible* if it never overestimates the *true cost* of reaching a goal from $s$, written $h^*(s)$ (and also called the *perfect* heuristic). A *pattern database (PDB)* heuristic $h^P$ is induced by a subset $P \subseteq \mathcal{V}$ of the variables of $\Pi$, called the *pattern*. $h^P(s)$ is defined as the perfect heuristic in the *projection* $\Pi_{|P}$ of $\Pi$ onto $P$, which can be computed by removing all occurrences of variables from $P$ in $\Pi$. PDBs are precomputed once by computing the optimal solution costs, $h^P(s^P)$, of all abstract states $s^P \in S^P$ in the abstract planning task $\Pi_{|P}$. During search, concrete states $s$ are mapped to abstract states $s^P$ using a perfect hash function (Sievers, Ortlieb, and Helmert 2012). This hash function is well-defined for all partial states $s^Q$ with $Q = vars(s) \supseteq P$, and we will abuse notation by writing $h^P(s^Q)$ for the heuristic computation of such states. PDBs grow exponentially in the number of included variables and thus single PDB heuristics alone typically do not provide enough guidance. Instead, state-of-the-art planners use different techniques for admissibly combining many PDB heuristics.

Let $H = \{h_1, \ldots, h_n\}$ be a set of arbitrary admissible heuristics $h_i$. We say that the heuristics in $H$ are *additive* if $h(s) = \sum_{i=1}^n h_i(s)$ is admissible. For non-additive heuristics, a trivial alternative for admissibly combining them is replacing the sum by the maximum (e.g. Holte et al. 2006).

A more advanced combination technique for the case that $h_i$ are PDB heuristics is due to Haslum et al. (2007) who define the *canonical PDB heuristic* which is based on the *disjoint additivity* of patterns underlying the PDBs. Two patterns $P_1$ and $P_2$ are disjoint-additive if there exist no $v_1 \in P_1$ and $v_2 \in P_2$ with $v_1, v_2 \in vars(\mathsf{eff}(a))$ for any action $a$ of the task. For a set of patterns $C$, also called *pattern collection*, the canonical PDB heuristic $h^C$ is defined as the maximum over the sums of PDBs induced by patterns in maximal disjoint-additive subsets, i.e., $h^C = \max_{A \in \mathcal{A}(C)} \sum_{P \in A} h^P$ where $\mathcal{A}(C)$ is the set of maximal disjoint-additive subsets of $C$. $h^C$ is admissible. There are other combination techniques for PDBs not covered here (e.g. Felner, Korf, and Hanan 2004; Pommerening, Röger, and Helmert 2013).

The most general general combination technique for a set of arbitrary admissible but non-additive heuristics $H$ is *cost partitioning* (Katz and Domshlak 2010; Pommerening et al. 2015). Cost partitioning computes each heuristic $h_i^{c_i}$ under a cost function $c_i$ different from the original cost function $c$ of the task. Then, $\sum_{i=1}^n h_i^{c_i}$ is admissible if $\sum_{i=1}^n c_i(a) \leq c(a)$ for all actions $a$ of the task. Since computing the *optimal* cost partitioning is usually infeasible in practice, in our experiments, we use the state-of-the-art *saturated cost partitioning (SCP)* (Seipp and Helmert 2018). In a nutshell, it considers the heuristics in $H$ in an arbitrary but fixed order. When computing $h_i^{c_i}$, it computes the *saturated costs* $\mathsf{scf}_i$, which are the minimum costs needed by the heuristic computation. They can be computed efficiently for abstraction heuristics by looping over all abstract transitions of the abstract state space. The costs not needed by $h_i$, $c_i - \mathsf{scf}_i$, are the costs $c_{i+1}$ available to the next heuristic $h_{i+1}$ and so on.

## Decoupled Search

Decoupled search decomposes a planning task by partitioning its variables into disjoint non-empty subsets, called a *factoring* $\mathcal{F} \subseteq 2^{\mathcal{V}}$. It imposes a structural requirement on the interaction between the *factors* $F \in \mathcal{F}$, a star topology, with a single *center factor* $C \in \mathcal{F}$ and an arbitrary number of *leaf factors* $\mathcal{L} := \mathcal{F} \setminus \{C\}$, such that the center can interact arbitrarily with the leaves, but leaves may only interact with each other if the center is involved as well. Formally, a factoring $\mathcal{F}$ is a *star factoring*, iff for all actions $a \in \mathcal{A}$ it holds that either there exists an $L \in \mathcal{L}$ such that $vars(\mathsf{pre}(a)) \subseteq C \cup L$ and $vars(\mathsf{eff}(a)) \subseteq L$, or $vars(\mathsf{eff}(a)) \cap C \neq \emptyset$. By imposing this structural requirement, decoupled search can efficiently handle cross-factor dependencies. Throughout this work, we will assume star factorings, and omit the "star".

Given a factoring $\mathcal{F}$ for a task $\Pi$, actions affecting $C$, i.e., with an effect on a variable in $C$, are called *center actions*, denoted $\mathcal{A}^C$, and those affecting a leaf are called *leaf actions*, denoted $\mathcal{A}^{\mathcal{L}}$; leaf actions of a particular leaf $L \in \mathcal{L}$ are denoted $\mathcal{A}^L$. A sequence of center actions applicable in $I$ in the projection of $\Pi$ onto $C$ is a *center path*; a sequence of $\mathcal{A}^L$-actions applicable in $I$ in the projection onto $L$ is a *leaf path*. A complete assignment to $C$ or to $L \in \mathcal{L}$ is called a *center state* or *leaf state*, respectively. $S^{\mathcal{L}}$ is the set of all leaf states and that of a particular leaf $L$ is denoted $S^L$.

A *decoupled state* $s^{\mathcal{F}}$ is a pair $\langle s^C(s^{\mathcal{F}}), \mathsf{prices}(s^{\mathcal{F}}) \rangle$ where $s^C(s^{\mathcal{F}})$ is a center state, and $\mathsf{prices}(s^{\mathcal{F}}) : S^{\mathcal{L}} \mapsto \mathbb{R}^{0+} \cup \{\infty\}$ is a *pricing function*, mapping each leaf state to a non-negative *price*. By $\pi^C(s^{\mathcal{F}})$ we denote the center path on which $s^{\mathcal{F}}$ was reached during search. The pricing function is maintained during decoupled search in a way so that the price of a leaf state $s^L$ is the cost of a cheapest leaf path that ends in $s^L$ and that is *compliant* with $\pi^C(s^{\mathcal{F}})$, i.e., that can be embedded into $\pi^C(s^{\mathcal{F}})$ such that the resulting action sequence is applicable in $I$ in the projection of $\Pi$ onto $C \cup L$. We denote the set of all decoupled states of $\Pi$ and $\mathcal{F}$

by $S^{\mathcal{F}}(\Pi)$. Decoupled search branches over center actions only, enumerating, for each leaf separately, the set of leaf states that can be reached in form of the pricing function. Every search algorithm can be employed on the decoupled state space (Gnad and Hoffmann 2018).

A decoupled state $s^{\mathcal{F}}$ represents a set of explicit states, its *member states*, which takes the form of a hypercube whose dimensions are the leaf factors $\mathcal{L}$. Formally, a state $s$ of $\Pi$ is a member state of a decoupled state $s^{\mathcal{F}}$, if $s[C] = s^C(s^{\mathcal{F}})$ and, for all leaves $L \in \mathcal{L}$, prices$(s^{\mathcal{F}})[s[L]] < \infty$. The *price* of $s$ in $s^{\mathcal{F}}$ is price$(s^{\mathcal{F}}, s) := \sum_{L \in \mathcal{L}}$ prices$(s^{\mathcal{F}})[s[L]]$. The *hypercube* of $s^{\mathcal{F}}$, denoted $[s^{\mathcal{F}}]$, is the set of all member states of $s^{\mathcal{F}}$. Hypercubes capture both the reachability and the price of the member states. For every member state $s$ we can construct an action sequence that starts in $I$ and ends in $s$ by augmenting $\pi^C(s^{\mathcal{F}})$ with cheapest-compliant leaf paths, i.e., the leaf paths that lead to the pricing function of $s^{\mathcal{F}}$.

A solution for a decoupled state $s^{\mathcal{F}}$ is a *decoupled plan*, i.e., a center path $\pi^C$ that starts in $s^{\mathcal{F}}$ and ends in a decoupled goal state $s_G^{\mathcal{F}}$ that contains a goal member state $s \in [s_G^{\mathcal{F}}]$ with $G \subseteq s$. The *augmented cost* of a decoupled plan is $c(\pi^C) + \min_{s \in [s_G^{\mathcal{F}}] \wedge G \subseteq s}$ price$(s_G^{\mathcal{F}}, s)$, which takes into account both the cost of the center actions from $s^{\mathcal{F}}$ to $s_G^{\mathcal{F}}$, as well as the price of the cheapest goal member state of $s_G^{\mathcal{F}}$. A decoupled plan for $s^{\mathcal{F}}$ is *augmented-optimal* if its augmented cost is minimal among all decoupled plans for $s^{\mathcal{F}}$.

In Figure 1 we illustrate two decoupled states from a simple logistics domain. The planning task has a truck $t$ that can *drive* between two locations $l, r$, and two packages $p_1, p_2$, that can be in the truck or at any of the two locations. Each package can be *loaded* into the truck if both are are the same position, or *unloaded* at the current truck position if it is in the truck. Initially, the truck and $p_1$ are at $l$ and $p_2$ is at $r$. We consider the factoring $\mathcal{F}$ where the truck is in the center $C = \{t\}$, and each package forms a leaf factor $L_i = \{p_i\}$. Decoupled search then branches over the *drive* actions, while *load* and *unload* are leaf actions. The left of Figure 1 depicts the initial decoupled state $I^{\mathcal{F}}$, with center state $\{t = l\}$ and where the pricing function assigns a price of 0 to the initial states of both leaves, i.e., $\{p_1 = l\}$ and $\{p_2 = r\}$, a price of 1 to $\{p_1 = t\}$ because the package can be loaded into the truck by applying a leaf action in the initial state (i.e., the leaf path $\langle load(p_1, l) \rangle$ is compliant with the empty center path of $I^{\mathcal{F}}$), and a price of $\infty$ for all remaining leaf states, which are not reachable. Applying $drive(r)$ in $I^{\mathcal{F}}$ results in the state on the right, with accordingly changed center state and where the pricing function is updated as shown. The initial state $I^{\mathcal{F}}$ contains two member states, namely $I = \{t = l, p_1 = l, p_2 = r\}$ and $s_1 = \{t = l, p_1 = t, p_2 = r\}$. The state $s^{\mathcal{F}}$ contains six member states, namely all combinations of leaf states with finite price together with the center state, e.g., $\{t = r, p_1 = 0, p_2 = t\} \in [s^{\mathcal{F}}]$.

Heuristics for decoupled search approximate the cost of augmented-optimal decoupled plans. Formally, a *decoupled heuristic* is a function $h_{\mathcal{F}} : S^{\mathcal{F}} \rightarrow \mathbb{R}^{0+} \cup \{\infty\}$. The *perfect decoupled heuristic* $h_{\mathcal{F}}^*$ returns the cost of an augmented-optimal decoupled plan for every decoupled state $s^{\mathcal{F}}$, or $\infty$

$$
\begin{array}{|ll|}
\hline
s^C(I^{\mathcal{F}}) = \{t = l\} & \\
\{p_1{=}l\}\rightarrow 0 & \{p_2{=}l\}\rightarrow\infty \\
\{p_1{=}t\}\rightarrow 1 & \{p_2{=}t\}\rightarrow\infty \\
\{p_1{=}r\}\rightarrow\infty & \{p_2{=}r\}\rightarrow 0 \\
\hline
\end{array}
\xrightarrow{drive(r)}
\begin{array}{|ll|}
\hline
s^C(s^{\mathcal{F}}) = \{t = r\} & \\
\{p_1{=}l\}\rightarrow 0 & \{p_2{=}l\}\rightarrow\infty \\
\{p_1{=}t\}\rightarrow 1 & \{p_2{=}t\}\rightarrow 1 \\
\{p_1{=}r\}\rightarrow 2 & \{p_2{=}r\}\rightarrow 0 \\
\hline
\end{array}
$$

Figure 1: Initial decoupled state $I^{\mathcal{F}}$ of the example and its successor state $s^{\mathcal{F}}$ via the $drive(r)$ center action.

if no such plan exists. As usual, a decoupled heuristic $h_{\mathcal{F}}$ is *admissible* if $h_{\mathcal{F}} \leq h_{\mathcal{F}}^*$.

Gnad and Hoffmann (2018) introduced a compilation that allows to compute (in principle) arbitrary heuristics for decoupled states. They construct a compiled planning task $\Pi_{L\$}$ in which the heuristic is forced to select and "buy" exactly one member state composed of a leaf state for each leaf:

**Definition 1.** *The buy-leaves compilation of a planning task* $\Pi = \langle \mathcal{V}, \mathcal{A}, I, G \rangle$ *and decoupled state* $s^{\mathcal{F}}$ *is the task* $\Pi_{L\$} = \langle \mathcal{V}_{L\$}, \mathcal{A}_{L\$}, s_{L\$}^{\mathcal{F}}, G_{L\$} \rangle$ *with cost function* $c_{L\$}$:

1. *The variables* $\mathcal{V}_{L\$}$ *include a new Boolean variable* bought$[L]$ *for every leaf* $L$, $\mathcal{V}_{L\$} := \mathcal{V} \cup \{\text{bought}[L] \mid L \in \mathcal{L}\}$. *For all leaf variables* $v \notin C$, *we add the new value* none *to* $\mathcal{D}(v)$.
2. *The initial state is* $s_{L\$}^{\mathcal{F}} := s^C(s^{\mathcal{F}}) \cup \{v = \text{none} \mid v \notin C\} \cup \{\text{bought}[L] = \bot \mid L \in \mathcal{L}\}$.
3. *The goal is* $G_{L\$} := G \cup \{\text{bought}[L] = \top \mid L \in \mathcal{L}\}$.
4. *The actions* $\mathcal{A}_{L\$}$ *are the previous ones* $\mathcal{A}$, *adding precondition* bought$[L] = \top$ *to* $a$ *whenever* $(vars(\text{pre}(a)) \cup vars(\text{eff}(a))) \cap L \neq \emptyset$. *We furthermore add, for every leaf* $L$, *and for every leaf state* $s^L \in S^L$ *where* prices$(s^{\mathcal{F}})[s^L] < \infty$, *a new "buy" action* $a[s^L]$ *with precondition* pre$(a[s^L]) := \{\text{bought}[L] = \bot\}$ *and effect* eff$(a[s^L]) := s^L \cup \{\text{bought}[L] = \top\}$.
5. *The cost function* $c_{L\$}$ *extends the previous one by setting* $c_{L\$}(a[s^L]) := $ prices$(s^{\mathcal{F}})[s^L]$ *for each new action* $a[s^L]$.

The leaf variables are assigned the value none initially to indicate that they "do not have a state yet". Before we can do anything relying on a leaf factor $L$, we have to "buy" (exactly) one of its states, at the price specified in the decoupled state $s^{\mathcal{F}}$ at hand. The price we pay in doing so accounts for $L$'s compliant path before $s^{\mathcal{F}}$; the classical plan obtained on $\Pi_{L\$}$ accounts for $L$'s compliant path behind $s^{\mathcal{F}}$.

Note that the goal in $\Pi_{L\$}$ forces the plan to buy a leaf state from *every* $L$, even if $L$ has no goal and would otherwise not be touched by any actions in the plan for $\Pi_{L\$}$. This is necessary because $L$ may have had to move *before* $s^{\mathcal{F}}$: we need to account for any costs incurred in $L$ in order to comply with the center path $\pi^C(s^{\mathcal{F}})$ leading to $s^{\mathcal{F}}$ in the first place.

Computing $h^*$ on the compilation results in $h_{\mathcal{F}}^*$ (Gnad and Hoffmann 2018), so the compilation is perfect in the limit. All existing decoupled heuristics are computed on the compilation. From now on, we write $h_{\mathcal{F}}$ to denote a decoupled heuristic based on the explicit heuristic $h$ (which we generally write without subscript). Throughout the paper, we will define several variants and ways of computing decoupled heuristics $h_{\mathcal{F}}$, which we denote by using additional sub-

scripts. For the decoupled heuristic obtained by computing the explicit heuristic $h$ on the compilation, we write $h_{\mathcal{F},\text{comp}}$.

## Decoupled Heuristics

The buy-leaves compilation can in principle be used to compute any heuristic. For PDB heuristics and abstractions in general, however, this does not make sense: one of the main advantages of PDBs is that the distance from every abstract state to the goal is *precomputed*, and the stored values are simply looked up every time the heuristic is evaluated on a state. The compilation, however, results in a different task for each decoupled state, and thus the PDB would need to be recomputed for each state evaluation. Moreover, the new bought$[L]$ variables should be considered during the pattern selection process, thus affecting the choice of patterns, effectively resulting in the computation of a different PDB. Simply ignoring these variables potentially enables abstract paths where multiple leaf states of one leaf are bought, selecting more than one member state, which leads to information loss.

To address these difficulties, we consider an alternative way of computing heuristics that does not rely on the compilation. We define the *explicit decoupled heuristic* as $h_{\mathcal{F},\text{ex}} = \min_{s \in [s^{\mathcal{F}}]} \text{price}(s^{\mathcal{F}}, s) + h(s)$. It evaluates an arbitrary explicit heuristic $h$ for a decoupled state $s^{\mathcal{F}}$ by minimizing the sum of price and heuristic over all member states. In this section, we analyze the properties of heuristics of this general form, where as the remainder of the paper deals with decoupled PDB heuristics and their combination in particular.

First, we show that computing the explicit decoupled *perfect heuristic* results in the perfect decoupled heuristic $h^*_{\mathcal{F}}$.

**Proposition 1.** *Let $s^{\mathcal{F}}$ be a decoupled state of a task $\Pi$ and factoring $\mathcal{F}$. Then $h^*_{\mathcal{F},ex} = h^*_{\mathcal{F}}$.*

*Proof.* Remember that for $h^*$, $h^*_{\mathcal{F}} = h^*_{\mathcal{F},\text{comp}}$ (Gnad and Hoffmann 2018), we hence show $h^*_{\mathcal{F},\text{ex}} = h_{\mathcal{F},\text{comp}}$ instead in the following. Let $\pi$ be an optimal plan in the buy-leaves compilation for $s^{\mathcal{F}}$, i.e., $c_{L\$}(\pi) = h^*_{\mathcal{F}}(s^{\mathcal{F}})$. Then, the subsequence $A[s^L]$ of actions $a[s^L]$ in $\pi$ corresponds to a member state $s \in s^{\mathcal{F}}$ that has been bought by $h^*_{\mathcal{F}}$. Note that the actions in $A[s^L]$ can be moved to the front of $\pi$, since all actions preceding $a[s^L] \in A[s^L]$ are neither preconditioned by nor affect $L \cup \{\text{bought}[L]\}$. Hence, the reordered plan $\pi'$ first buys the member state $s$, then solves $s$. Converting the suffix of $\pi'$ back to the original task by removing all bought$[L]$-preconditions results in a plan for $s[\mathcal{V}]$ for $\Pi$. Since $h^*_{\mathcal{F}}$ returns the cost of an optimal plan for $\Pi_{L\$}$, there is no member state $s' \in [s^{\mathcal{F}}]$ where $\text{price}(s^{\mathcal{F}}, s') + h^*(s') < h^*_{\mathcal{F}}(s^{\mathcal{F}})$.

For the other direction, observe that we can transform any solution for $\min_{s \in [s^{\mathcal{F}}]} \text{price}(s^{\mathcal{F}}, s) + h^*(s)$ with underlying optimal plan $\pi$ for $s$ into a plan for $\Pi_{L\$}$ with the same cost. Since the sum of the price of $s$ and an optimal plan for $s$ is minimal, there does not exist a cheaper solution of $\Pi_{L\$}$.  □

For any admissible heuristic $h$, since $h$ lower bounds $h^*$, computing $h_{\mathcal{F},\text{ex}}$ is a lower bound for $h^*_{\mathcal{F}}$:

**Corollary 1.** *Let $h$ be an admissible heuristic. Then $h_{\mathcal{F},ex}$ is admissible.*

While both $h_{\mathcal{F},\text{ex}}$ and the buy-leaves compilation result allow us to compute any admissible heuristic in decoupled search, a relevant question is which of them is more informative. In fact, even though the buy-leaves compilation is perfect in the limit, for heuristics based on common relaxations there can be an additional information loss inherent to the compilation. For delete-relaxation heuristics, for example, this is because an arbitrary set of leaf states can be bought for each leaf $L$, because the "delete" effect bought$[L] = \bot$ is ignored. $h_{\mathcal{F},\text{ex}}$ overcomes this information loss by explicitly enumerating and computing the heuristic for all member states of a decoupled state. An interesting observation is that for any *consistent* heuristic $h$, $h_{\mathcal{F},\text{ex}}$ strictly dominates $h_{\mathcal{F},\text{comp}}$. To prove this, we need an additional requirement on the employed heuristic:

**Definition 2.** *Let $\Pi$ be a planning task, $\mathcal{F}$ a factoring for $\Pi$, and $s^{\mathcal{F}} \in S^{\mathcal{F}}(\Pi)$ a decoupled state. Let further $\Pi_{L\$}$ be the buy-leaves compilation for $s^{\mathcal{F}}$, and $h, h^{L\$}$ be two heuristic functions, where $h$ is defined on $\Pi$ and $h^{L\$}$ on $\Pi_{L\$}$.*

*Then the pair of heuristics $\langle h, h^{L\$} \rangle$ is buy-leaves agnostic iff for all $s \in S(\Pi) : h(s) = h^{L\$}(s \cup \{\text{bought}[L] = \top \mid L \in \mathcal{L}\})$.*

It is reasonable to assume that heuristics are buy-leaves agnostic. After applying a buy action $a[s^L]$ for each leaf in the compilation, the resulting states $s^{L\$}$ are exactly the member states $s \in [s^{\mathcal{F}}]$ extended by bought$[L] = \top$ facts for all leaves. In every descendant of such a $s^{L\$}$ the additional precondition bought$[L] = \top$ of all original actions $a \in \mathcal{A}$ that were adapted in the compilation is always fulfilled, and none of the new actions $a[s^L]$ is ever applicable. Therefore, the reachable state spaces behind $s$ and $s^{L\$}$ are isomorphic and the successor states are identical except for the bought$[L] = \top$ facts. The $h^{\max}$ heuristic (Bonet and Geffner 2001), for example, is buy-leaves agnostic.

**Proposition 2.** *Let $h, h^{L\$}$ be two consistent and buy-leaves agnostic heuristics. Then, for every decoupled state $s^{\mathcal{F}}$ it holds that $h_{\mathcal{F},ex}(s^{\mathcal{F}}) \geq h_{\mathcal{F},comp}(s^{\mathcal{F}})$.* [1]

Cases where the inequality is strict, i.e., where $h_{\mathcal{F},\text{ex}}(s^{\mathcal{F}}) > h_{\mathcal{F},\text{comp}}(s^{\mathcal{F}})$, can be constructed, too. A PDB heuristic that is defined on the same subset of variables for both $\Pi$ and $\Pi_{L\$}$, i.e., that ignores the bought$[L]$ variables, is able to use multiple buy actions for a leaf $L$ and thereby combine cheap leaf facts of $L$, leading to lower overall cost.

## Pattern Databases for Decoupled Search

For any explicit heuristic, we can compute its decoupled counterpart $h_{\mathcal{F},\text{ex}}$ by enumerating all member states. However, this is in general not a good idea, as there is an exponential number of member states in $[s^{\mathcal{F}}]$ so this foregoes the advantage of the compact decoupled state representation. When computing PDB heuristics, however, we can take advantage of the fact that the heuristic value of a PDB only depends on a subset of the variables/leaf factors.

Let $\Pi$ be a planning task with variables $\mathcal{V}$, and let $P \subseteq \mathcal{V}$ be a pattern. We write $\mathcal{L}_{h^P}$ to denote the set

---

[1] Full proofs are published online in the extra material (Sievers, Gnad, and Torralba 2022a).

of leaves *affected* by a PDB heuristic $h^P$, i.e., $\mathcal{L}_{h^P} :=$ $\{L \in \mathcal{L} \mid L \cap P \neq \emptyset\}$. We define the *price of an abstract state* $s^P \in S^P$ as $\text{price}(s^{\mathcal{F}}, s^P) :=$ $\sum_{L \in \mathcal{L}_{h^P}} \min_{s^L \in S^L, s^P[L] \subseteq s^L} \text{prices}(s^{\mathcal{F}})[s^L]$, i.e., for each affected leaf we select the reached leaf state with minimum price that is projected to the leaf part of $s^P$ and sum-up those minimum prices. With this, we have a new approach to compute a PDB heuristic for a decoupled state, which we call *decoupled PDB*: $\text{dPDB}(h^P, s^{\mathcal{F}}) = \min_{s^P \in S^P} \text{price}(s^{\mathcal{F}}, s^P) + h^P(s^P)$

Note that this can be computed in polynomial time in the number of abstract states $|S^P|$, without enumerating all member states. In practice, the number of abstract states can sometimes be larger than the number of member states (with finite price) in $s^{\mathcal{F}}$. Therefore, instead of iterating over all abstract states, one can iterate directly over those such that $\text{price}(s^{\mathcal{F}}, s^P) < \infty$. To do so, we can, for each leaf separately, compute the set of reached abstract leaf states, i.e., the set $\{s^L[P] \mid \text{prices}(s^{\mathcal{F}})[s^L] < \infty\}$, and their minimum price $\text{minprice}(s^L[P]) = \min_{s^L \in S^L, s^L[P] \subseteq s^L} \text{prices}(s^{\mathcal{F}})[s^L]$. In a second step, we multiply out these sets across leaves, which, augmented with the abstract center state $s^C(s^{\mathcal{F}})[P]$, results in full abstract states $s^P \in S^P$, and minimize the sum of their price and heuristic value. The number of abstract states generated thereby is upper-bounded by the number of member states of $s^{\mathcal{F}}$ and the number of abstract states of $P$, but is typically much smaller than both.

The following proposition shows that the heuristic values computed by dPDB are equivalent to the explicit decoupled PDB heuristic $h^P_{\mathcal{F}, \text{ex}}$, except for the missing inclusion of leaf prices of leaves unaffected by $P$.

**Proposition 3.** *Let $\Pi$ be a planning task with factoring $\mathcal{F}$ and $h^P$ a PDB heuristic. Then $h^P_{\mathcal{F}, \text{ex}}(s^{\mathcal{F}}) = \text{dPDB}(h^P, s^{\mathcal{F}}) + \sum_{L \in \mathcal{L} \setminus \mathcal{L}_{h^P}} \min_{s^L \in S^L} \text{prices}(s^{\mathcal{F}})[s^L]$.*

*Proof.* Let $s$ be the member state that minimizes $h^P_{\mathcal{F}, \text{ex}}(s^{\mathcal{F}})$. Then: $h^P_{\mathcal{F}, \text{ex}}(s^{\mathcal{F}}) = \text{price}(s^{\mathcal{F}}, s) + h^P(s) = \sum_{L \in \mathcal{L}} \text{prices}(s^{\mathcal{F}})[s[L]] + h^P(s[P]) = \sum_{L \in \mathcal{L}_{h^P}} \text{prices}(s^{\mathcal{F}})[s[L]] + h^P(s[P]) + \sum_{L \in \mathcal{L} \setminus \mathcal{L}_{h^P}} \text{prices}(s^{\mathcal{F}})[s[L]] = \text{dPDB}(h^P, s^{\mathcal{F}}) + \sum_{L \in \mathcal{L} \setminus \mathcal{L}_{h^P}} \text{prices}(s^{\mathcal{F}})[s[L]]$. The claim follows with the observation that $s$ minimizes $h^P_{\mathcal{F}, \text{ex}}(s^{\mathcal{F}})$. Since $h^P$ is independent of the assignments to variables in $\mathcal{V} \setminus P$, there are no other leaf states of any $L \in \mathcal{L} \setminus \mathcal{L}_{h^P}$ with a price lower than $\text{prices}(s^{\mathcal{F}})[s[L]]$. $\square$

In the remainder of this section, we discuss the relationship of $h^P_{\mathcal{F}, \text{ex}}$ and PDB heuristics computed on the compilation. Our first result shows that, given an arbitrary PDB heuristic $h^P$, we can construct a heuristic for the buy-leaves compilation that is just as informative as $h^P_{\mathcal{F}, \text{ex}}$, by incorporating all bought$[L]$ variables into the pattern.

**Proposition 4.** *Let $\Pi$ be a planning task with factoring $\mathcal{F}$, and $h^P$ a PDB heuristic for $\Pi$. There exists a heuristic $h^{P'}_{\mathcal{F}, \text{comp}}$ for the buy-leaves compilation $\Pi_{L\$}$ such that for*

*all decoupled states $s^{\mathcal{F}}$: $h^{P'}_{\mathcal{F}, \text{comp}}(s^{\mathcal{F}}) = h^P_{\mathcal{F}, \text{ex}}(s^{\mathcal{F}})$.*

Note that the construction in the proof leads to an increase in the size of the PDB's abstract state space that is exponential in the number of leaves. Our second result shows that not adapting the pattern to the compilation in general results in an information loss.

**Proposition 5.** *$h^P_{\mathcal{F}, \text{comp}}(s^{\mathcal{F}}) \leq h^P_{\mathcal{F}, \text{ex}}(s^{\mathcal{F}})$.*

The previous two propositions, while not very relevant in practice since we never compute PDB heuristics on the buy-leaves compilation, indicate an issue that we will face when considering additive PDB collections. Decoupled heuristics take into account the pricing function of the given decoupled state, and it is important to do so to incorporate all available information to get good estimates. Let $P_1, P_2$ be two disjoint-additive patterns. Then, taking the perspective of the buy-leaves compilation, we need to include the bought$[L]$ variables into both patterns to not lose information. But then the patterns are no longer disjoint-additive, so we cannot simply sum-up their estimates admissibly. We will propose solutions to this issue in the next section.

## Multiple PDBs for Decoupled Search

As the size of PDBs grows exponentially in the number of variables included in their pattern, planners typically compute a set of small PDB heuristics and combine them admissibly with one of the techniques mentioned in the background section. In the following, we assume that we are provided with a set $\mathcal{H}$ of additive sets $H$ of admissible PDB heuristics. In the spirit of the canonical PDB heuristic, we admissibly combine the heuristics as $h^{\mathcal{H}}(s) = \max_{H \in \mathcal{H}} \sum_{h \in H} h(s)$. Note that this definition covers both the canonical PDB heuristic (in which case $\mathcal{H} = \mathcal{A}(C)$ for the set of patterns $C$ underlying all heuristics occurring in $\mathcal{H}$, and heuristics in $\mathcal{H}$ can be part of several additive sets $H$) and arbitrarily cost-partitioned heuristics (in which case multiple heuristics in $\mathcal{H}$ can be computed over the same pattern, but most likely using a different cost function). In the following we drop the superscript $P$ of PDB heuristics.

Besides $h^{\mathcal{H}}$, we also consider two special cases:

- $\mathcal{H}$-sum: $|\mathcal{H}| = 1$, the heuristic is a set of additive PDBs.
- $\mathcal{H}$-max: $|H| = 1$ for all $H \in \mathcal{H}$ and the heuristic is simply the maximum of a set of PDBs.

To evaluate $h^{\mathcal{H}}$ for decoupled search, the straightforward way is to use the explicit decoupled heuristic (cf. section "Decoupled Heuristics"): $h^{\mathcal{H}}_{\mathcal{F}, \text{ex}}(s^{\mathcal{F}}) = \min_{s \in [s^{\mathcal{F}}]} \text{price}(s^{\mathcal{F}}, s) + \max_{H \in \mathcal{H}} \sum_{h \in H} h(s)$.

In our previous example, assuming three PDBs with patterns $P_1 = \{t, p_1\}$, $P_2 = \{p_2\}$, and $P_3 = \{p_1, p_2\}$, where $P_1$ and $P_2$ are additive, so $\mathcal{H} = \{\{h^{P_1}, h^{P_2}\}, \{h^{P_3}\}\}$, and that the goal is $G = \{p_1 = r, p_2 = l\}$, the heuristic value of the initial decoupled state $I^{\mathcal{F}}$ is obtained as follows:

$$h^{\mathcal{H}}_{\mathcal{F}, \text{ex}}(I^{\mathcal{F}}) =$$
$$\min(\text{price}(I^{\mathcal{F}}, I) + \max(h^{P_1}(I) + h^{P_2}(I), h^{P_3}(I)),$$
$$\text{price}(I^{\mathcal{F}}, s_1) + \max(h^{P_1}(s_1) + h^{P_2}(s_1), h^{P_3}(s_1))$$
$$= \min(0 + \max(3 + 2, 4), 1 + \max(2 + 2, 3)) = 5$$

Besides this exact way, we suggest several approximations in the following, and discuss their complexity.

## Naïve Combination

As explained in the previous section, the value of a single PDB can be computed efficiently for decoupled states. Therefore, a natural and naïve approximation to compute decoupled $h^{\mathcal{H}}$ is to combine the individual estimates from the decoupled PDBs: $h^{\mathcal{H}}_{\mathcal{F},\text{naïve}}(s^{\mathcal{F}}) = \max_{H \in \mathcal{H}} \sum_{h \in H} h_{\mathcal{F},\text{ex}}(s^{\mathcal{F}})$.

However, such a simple combination is not valid, as it not only induces information loss, but also makes the sum inadmissible. The underlying reason for the information loss is that each decoupled PDB could select a different member state which minimizes the heuristic value, whereas $h^{\mathcal{H}}_{\mathcal{F},\text{ex}}$ ensures that the same member state is chosen by all PDBs. This happens even in $\mathcal{H}$-max, where we are only taking the maximum out of a set of PDBs.

**Proposition 6.** $h^{\mathcal{H}\text{-max}}_{\mathcal{F},\text{naïve}} \leq h^{\mathcal{H}\text{-max}}_{\mathcal{F},\text{ex}}$ *and there are cases where the inequality is strict.*

*Proof Sketch.* We show that $h^{\mathcal{H}\text{-max}}_{\mathcal{F},\text{naïve}}$ and $h^{\mathcal{H}\text{-max}}_{\mathcal{F},\text{ex}}$ differ only on the order of the min and max operations, and the claim then follows with the max-min inequality. □

Note that the same example can be used to show that the sum of multiple heuristics also incurs an information loss. Furthermore, the sum is not admissible, as each heuristic may include the price of leaf states from the same leaf.

**Proposition 7.** $h^{\mathcal{H}\text{-sum}}_{\mathcal{F},\text{naïve}}$ *is not admissible.*

*Proof.* Consider the sum of two PDB heuristics $h_1$ and $h_2$ computed for a decoupled state $s^{\mathcal{F}}$ with two member states $[s^{\mathcal{F}}] = \{s_1, s_2\}$. Assume $h_1(s_1) = h_2(s_1) = 1$, $h_1(s_2) = h_2(s_2) = \infty$, and $h^*(s_1) = 2$. Furthermore $\text{price}(s^{\mathcal{F}}, s_1) = 10$ and $\text{price}(s^{\mathcal{F}}, s_2) = 0$. Then, $h^{\mathcal{H}}_{\mathcal{F},\text{naïve}}(s^{\mathcal{F}}) = 11 + 11 = 22 > h^*_{\mathcal{F}}(s^{\mathcal{F}}) = 12$. □

## Hardness of the Explicit Decoupled Additive PDBs

After ruling out the naïve computation, we consider to compute $h^{\mathcal{H}}_{\mathcal{F},\text{ex}}$. However, computing this turns out to be an **NP**-complete problem, as it requires finding the member state that minimizes the heuristic value for the entire PDB collection. This is also true for the special cases $\mathcal{H}$-max and $\mathcal{H}$-sum. In the following, we prove this result for $\mathcal{H}$-sum and explain how it transfers to $\mathcal{H}$-max and detecting dead ends.

**Definition 3** (Decoupled Additive PDBs Problem). *Given a planning task $\Pi$ with factoring $\mathcal{F}$, a reachable decoupled state $s^{\mathcal{F}}$, a bound $B \in \mathbb{R}^+ \cup \{\infty\}$, and a set of additive PDBs $\mathcal{H}$, decide whether $h^{\mathcal{H}\text{-sum}}_{\mathcal{F},\text{ex}}(s^{\mathcal{F}}) < B$.*

We require $s^{\mathcal{F}}$ to be reachable to show that this is not only hard for decoupled states with arbitrarily defined pricing functions, but that such states can be encountered during the search. Also, note that we consider $\mathcal{H}$ as a set of PDBs, and not patterns. Therefore, the size of the input is proportional to the number of abstract states in the pattern collection. Thus, the corresponding problem for explicit search

can be computed in polynomial time simply by looking up the value of $s$ in all PDBs and adding the corresponding values. However, the same is not true in decoupled search:

**Theorem 1.** *Decoupled Additive PDBs is **NP**-complete.*

*Proof Sketch. Membership:* Guess a member state $s \in [s^{\mathcal{F}}]$, test in polynomial time if $\text{price}(s^{\mathcal{F}}, s) + h^{\mathcal{H}\text{-sum}}(s) < B$.

*Hardness:* Reduction from 3-SAT. Given any 3-CNF formula $\phi$ and clauses $\{C^1, \ldots, C^m\}$, we construct a planning task and decoupled state $s^{\mathcal{F}}$ s.t. $h^{\mathcal{H}\text{-sum}}_{\mathcal{F},\text{ex}}(s^{\mathcal{F}}) = 0$ if $\phi$ is satisfiable, and $h^{\mathcal{H}\text{-sum}}_{\mathcal{F},\text{ex}}(s^{\mathcal{F}}) = \infty$ otherwise. Our construction has a leaf $L_i$ per clause $C^i$, that represents assignments over the 3 propositions in $C^i$. In $s^{\mathcal{F}}$, each member state corresponds to assignments satisfying all clauses, but assignments to each proposition may be different in each leaf. Then, the PDB collection contains a PDB per pair of clauses, which identifies as dead-end any state containing an inconsistent assignment among those clauses. Therefore, there exists a member state $s \in [s^{\mathcal{F}}]$ for which $\text{price}(s^{\mathcal{F}}, s) + h^{\mathcal{H}\text{-sum}}(s) < \infty$ iff $\phi$ is satisfiable. □

Note that in our construction action costs are irrelevant, so exactly the same argument applies to $h^{\mathcal{H}\text{-max}}_{\mathcal{F},\text{ex}}$, and to the problem of deciding whether $s^{\mathcal{F}}$ is a dead end.

**Corollary 2.** $h^{\mathcal{H}}_{\mathcal{F},\text{ex}}$ *cannot be computed in polynomial time in the size of the leaf state spaces in $\mathcal{F}$ and the abstract state spaces of PDBs in $\mathcal{H}$ unless $P = NP$.*

## Practical Implementation

Solving an **NP**-complete problem for every decoupled state evaluation may seem completely impractical. Yet, as shown in the evaluation, our algorithm that solves the problem exactly is competitive with the polynomial approximations introduced later in this section. It computes $h^{\mathcal{H}}_{\mathcal{F},\text{ex}}$ by enumerating the member states. If done in a clever way, many member states can possibly be skipped, namely if their partially computed heuristic is higher than the current best estimate.

The algorithm, shown in Algorithm 1, recursively (via index $i$) enumerates the member states of a decoupled state. This is done by incrementally constructing a member state (parameter $s$), from the center state of $s^{\mathcal{F}}$ and all combinations of reached leaf states. Along with $s$, we incrementally compute its price $p_s$ in $s^{\mathcal{F}}$. Furthermore, we compute the heuristic value $h_s = h^{\mathcal{H}}(s)$. This heuristic evaluation can be done only at the bottom of the recursion in the "else" case, or—as done in the algorithm—incrementally as soon as a heuristic can be computed on the partial state $s \cup s^L$. The incremental computation uses a restriction $H_{\leq i}$ of $H$ that contains only those $h$ whose affected leaves have already been treated during the recursion, formally $H_{\leq i} = \{h \in H \mid \mathcal{L}_h \subseteq \{L_1, \ldots, L_i\}\}$. The final heuristic value $h_{\min}$ (a global variable) is initialized to $\infty$ and updated at the bottom of the recursion.

The incremental heuristic computation of the algorithm allows *pruning*, i.e., skipping a recursive call, if the lower bound on the heuristic value of $s$, i. e., $p_s + p_{s^L} + h_s$, already exceeds the current minimum value $h_{\min}$. Note that this includes cases where $h_s$ is a dead end. (If all member states

---

**Algorithm 1:** Recursive computation of $h_{\mathcal{F},\text{ex}}^{\mathcal{H}}(s^{\mathcal{F}})$

---

**Input:** Heuristics $\mathcal{H} = \{H_1, \ldots, H_n\}$, dec. state $s^{\mathcal{F}}$
**Output:** $h_{\mathcal{F},\text{ex}}^{\mathcal{H}}(s^{\mathcal{F}})$

1  $h_{\min} \leftarrow \infty$
2  $\mathcal{L} = \langle L_1, \ldots, L_n \rangle$             // order leaves
3  enumerate$(0, s^C(s^{\mathcal{F}}), 0)$
4  **return** $h_{\min}$

5  **def** enumerate$(i, s, p_s)$:
6      **for** $s^L \in S^{L_i} : \text{prices}(s^{\mathcal{F}})[s^L] < \infty$ **do**
7          $p_{s^L} \leftarrow \text{prices}(s^{\mathcal{F}})[s^L]$
8          $h_s \leftarrow \max_{H \in \mathcal{H}} \sum_{h \in H_{\leq i}} h(s \cup s^L)$
9          **if** $p_s + p_{s^L} + h_s \geq h_{\min}$ **then**
10             **continue**                // pruning
11         **if** $i < n$ **then**
12             enumerate$(i+1, s \cup s^L, p + p_{s^L})$
13         **else**
14             $h_{\min} \leftarrow \min(h_{\min}, p + p_{s^L} + h_s)$

---

are dead ends, then $h_{\min} = \infty$ is never changed). Since the order of leaf factors influences the pruning by determining at which level in the recursion an $h \in H$ can be computed (leaves that are not affected by any heuristic can be skipped), we experimentally evaluate two different such orders.

We also considered encoding the computation of $h_{\mathcal{F},\text{ex}}^{\mathcal{H}}$ as MIP, but found Algorithm 1 to always perform better.

### Polynomial-time Approximations

Given that the admissible combination of multiple PDBs is **NP**-hard in general, we propose polynomial-time approximations of $h_{\mathcal{F},\text{ex}}^{\mathcal{H}}$. An important part is to approximate the maximum over $\mathcal{H}$ by considering each additive subset $H$ independently. That is, we swap the $\min$ and $\max$ operators: $\max_{H \in \mathcal{H}} \min_{s \in [s^{\mathcal{F}}]} \text{price}(s^{\mathcal{F}}, s) + \sum_{h \in H} h(s)$. By Proposition 6, this is a lossy but admissible approximation of $h_{\mathcal{F},\text{ex}}^{\mathcal{H}}$.

For each additive subset $H$, we consider two special cases for which the estimate can be computed in polynomial time by imposing restrictions on the PDBs considered by $H$.

**Case 1: leaf-disjoint PDBs**   Let $H = \{h_1, \ldots, h_n\}$ be a set of additive PDBs. We say that two heuristics $h_1, h_2$ are *leaf-disjoint* if they do not affect the same leaves, i.e., $\mathcal{L}_{h_1} \cap \mathcal{L}_{h_2} = \emptyset$. If all heuristics in $H$ are pairwise leaf-disjoint, we can simply sum up the estimates and obtain $h_{\mathcal{F},\text{ex}}^{\mathcal{H}\text{-sum}}$:

**Proposition 8.** *Let $\mathcal{H} = \{H\}$, where $H$ is a set of additive PDBs. If all heuristics in $H$ are pairwise leaf-disjoint, then $h_{\mathcal{F},\text{naïve}}^{\mathcal{H}\text{-sum}} = h_{\mathcal{F},\text{ex}}^{\mathcal{H}\text{-sum}}$.*

*Proof Sketch.* We can move the sum in $h_{\mathcal{F},\text{ex}}^{\mathcal{H}\text{-sum}}$ outward because all PDBs are leaf-disjoint, so we can minimize the sum of prices and heuristic separately for each $h$ on the subset of leaves affected only by $h$, i.e., over the partial member states induced by $\mathcal{L}_h$. $\qquad\square$

Note that the computation is also possible in polynomial time because it only adds estimates of separately computed

single-PDB heuristics, which can be done in polynomial-time as discussed in the previous section.

The "leaf disjointness" of PDBs in $H$ can be enforced easily both when using the CPDB and the SCP heuristics. For the former, we do not consider two patterns to be disjoint additive if they have any leaf variable in common, so the maximal disjoint-additive subsets are restricted. For the latter, we skip, during computation, any PDB in $H$ if it affects a leaf affected by any non-skipped PDB ordered earlier in the SCP computed over $H$.

**Case 2: single-leaf PDBs**   The second case limits each PDB to affect at most a single leaf. This means we can partition $H$ into subsets $H = H_{L_1} \cup \cdots \cup H_{L_n}$ where heuristics in each $H_{L_i}$ affect only leaf $L_i$. Then the sum over $H_{L_i}$ corresponds to the leaf-disjoint case detailed before, so it is admissible. We can compute the heuristic for each $H_{L_i}$ efficiently because it suffices to consider only a single leaf $L_i$. The heuristic value of $H$ is defined as:

$$\sum_{L \in \mathcal{L}} \min_{s^L \in S^L} \text{prices}(s^{\mathcal{F}})[s^L] + \sum_{h \in H_L} h(s^C(s^{\mathcal{F}}) \cup s^L) \quad (1)$$

Here, $s^C(s^{\mathcal{F}}) \cup s^L$ is a partial state defined on at least all variables of the pattern of $h$ and can thus be used by heuristic $h$. Computing Equation 1 is polynomial-time because in contrast to the explicit decoupled heuristic, it only requires iterating over all reached leaf states of each leaf once. Since all heuristics only affect a single leaf, the sum over all leaves is admissible and corresponds to $h_{\mathcal{F},\text{ex}}^{\mathcal{H}\text{-sum}}$:

**Proposition 9.** *Let $\mathcal{H} = \{H\}$, where $H$ is a set of additive PDBs each affecting at most one leaf. Then Equation 1 equals $h_{\mathcal{F},ex}^{\mathcal{H}\text{-sum}}$.*

*Proof Sketch.* As each $h \in H$ affects only a single leaf, we can compute $h_{\mathcal{F},\text{ex}}^{\mathcal{H}\text{-sum}}$ separately per leaf and sum up these minimum values across leaves. This is possible because each $h \in H_L$ can be computed using only the center state $s^C(s^{\mathcal{F}})$ and a leaf state of $L$, as all variables of the pattern are part of such a partial state. $\qquad\square$

## Experiments

We implemented our techniques in the Fast Downward planner (Helmert 2006). In particular, we build upon the code base for decoupled search (Gnad 2021) and adapted code from the Scorpion planner (Seipp 2018) to compute SCPs.

We compute single PDB heuristics using the simple greedy algorithm of Fast Downward. For the CPDB heuristic, we compute patterns using the hill-climbing (HC) approach by Haslum et al. (2007) for 900s. For the SCP heuristic, we compute patterns by running the multiple CEGAR algorithm by Rovner, Sievers, and Helmert (2019) for 100s, systematically generating all patterns up to size 2 (Pommerening, Röger, and Helmert 2013) for 100s, and running HC for 500s. We then compute a set of diverse heuristic orders for these PDBs for 200s, each optimized using dynamic greedy ordering for 2s (Seipp, Keller, and Helmert 2020).

We use EXP to denote computing the explicit decoupled heuristic using Algorithm 1. For single PDBs, we also evaluate the efficient computation of decoupled PDBs (dPDB).

For combining multiple PDBs, we evaluate the leaf disjoint (LD) and single leaf (SL) approximations. For CPDB-LD, we adapt HC to use CPDB-LD to optimize the pattern collection for the right heuristic. With SCP-LD, there is nothing analogous we could do in the pattern generation, but we modify the order optimization of the SCP computation to use SCP-LD computation. For the SL approximation, we restrict all pattern generation algorithms to single-leaf patterns.

As baseline, we consider explicit $A^*$ search (E). Besides the "pure" variant (pE), we consider a variant that computes the factoring (fE) which allows evaluating the impact of restricting pattern generation. For decoupled search (D), we consider three types of factorings: forks (F) and inverted forks (IF) (Gnad, Poser, and Hoffmann 2017), and LP-based strategies for general star factorings that maximize leaf mobility (MM) (Schmitt, Gnad, and Hoffmann 2019). We also include results for LM-cut (Helmert and Domshlak 2009).

We run all planners on the benchmarks from the optimal sequential track of all IPCs, a set comprised of 1827 tasks over 65 domains. We conduct experiments on Intel Xeon Silver 4114 CPUs using Downward Lab (Seipp et al. 2017) to limit each planner to 1800s and 3.5 GiB. We evaluate planners by the number of state expansions, search time in seconds (excluding preprocessing time of heuristics), and the number of solved tasks (coverage). We never mix results for the three factoring methods, and for each method, we only consider tasks in which it finds a non-trivial factoring. The additional material (Sievers, Gnad, and Torralba 2022a) also contains results on the Autoscale 21.11 benchmarks (Torralba, Seipp, and Sievers 2021) and full per-domain coverage results. The code, benchmarks and experimental data are published online (Sievers, Gnad, and Torralba 2022b).

First, we focus our attention to single PDB heuristics to compare EXP and dPDB for computing decoupled PDBs. Figure 2a plots search time of EXP against dPDB for factorings F and MM; with IF, there are nearly no differences. We observe that the efficient computation can indeed lead to a significant speed-up and comes at no risk. Therefore, we always use dPDB when combining multiple PDBs with the LD approximation (which needs to compute decoupled PDBs) in the following.

Next, we report results for combining multiple PDBs using the CPDB and SCP heuristics. Table 1 shows coverage. First, consider the results for explicit search in the first four columns. Differences between pE and fE are only due to fE computing the factoring, and we compare fE against LD and SL to evaluate the impact of restricting PDBs. We see that the restriction leads to a loss in coverage most of times, but it increases for some combinations. Investigating this further for CPDBs, we found that the restriction to LD has *no* effect on heuristic quality for all three factorings. We conjecture that this is due to HC only generating patterns containing a single goal variable and its predecessors in the causal graph, thus the goal determines which leaves a pattern affects. The restriction to SL always results in worse heuristics with the exception of a few tasks. When looking at SCPs, the picture changes: LD has a strongly negative impact, which is, however, mostly attributed to the two domains miconic and wookworking. With the SL restriction, coverage decreases

| | | explicit search (E) | | | | decoupled search (D) | | | | | |
| | | | | | | EXP | | | | | |
| | | p | f | LD | SL | NP | NI | FD | AFF | LD | SL |
|---|---|---|---|---|---|---|---|---|---|---|---|
| CPDB | F | 201 | 201 | 201 | 183 | 182 | 188 | 190 | 194 | **211** | 204 |
| | IF | **192** | 191 | 191 | 174 | 179 | 186 | 185 | 186 | 189 | 171 |
| | MM | **672** | 644 | 644 | 618 | 573 | 587 | 590 | 591 | 605 | 585 |
| SCP | F | 283 | 284 | 206 | 293 | 206 | 208 | 208 | 212 | 210 | **304** |
| | IF | **226** | 221 | 208 | 206 | 195 | 200 | 201 | 200 | 196 | 193 |
| | MM | **784** | 749 | 662 | 743 | 596 | 603 | 630 | 628 | 607 | 707 |

Table 1: Coverage of CPDBs and SCPs with the three factorings fork (F), inverted fork (IF), and general star (MM). Left block: explicit search (p: pure, f: with factoring though not using it, LD/SL: leaf-disjoint/single leaf restricted patterns). Right block: decoupled search with explicit heuristic computation with Algorithm 1 (EXP) (without pruning (NP) and with pruning with non-incremental computation (NI) and incremental computation with leaf factor order corresponding to Fast Downward's variable order (FD) and preferring leaves affected by many PDBs (AFF)) and with the polynomial-time heuristic approximations LD and SL.

with IF and MM and increases with F. A closer look at heuristic quality, as demonstrated by expansion plots in Figure 2b comparing unrestricted SCP against LD and SL, reveals that restricting PDBs can be both beneficial and bad. LD reduces the number of heuristics that SCP can combine, which potentially removes good heuristics, but which also allows later ordered heuristics to contribute more to the SCP, which can be beneficial if these heuristics are better than the skipped ones. SL restricts the generated heuristics and therefore has unpredictable effects on techniques that combine these heuristics. Generally speaking, we see that SCP does not suffer from the SL restriction as much as CPDB does, which confirms that SCP is a stronger technique for combining arbitrary sets of PDBs than CPDBs, which rely on the simpler disjoint-additive criterion.

We now turn to decoupled search. In the first four columns of that block, we report results for the explicit decoupled heuristic computation (EXP). The first column is without pruning (NP), the others with pruning, using the non-incremental (NI) variant as well as two incremental ones with two different (precomputed) leaf factor orderings: FD uses the order of leaf factors in Fast Downward; AFF orders leaves by decreasing number of PDBs affecting them.[2] We observe that pruning always improves coverage, and both incremental variants fare better than the non-incremental one, with the AFF ordering having a clear edge in particular with SCP. Since AFF prefers enumerating leaf states of leaves which are affected by many PDBs, we can partially compute many heuristic values early on, allowing for more pruning.

We next compare EXP to the two polynomial approximations. For CPDBs, using SL mostly decreases coverage,

---

[2]We also experimented with a simple *dynamic* order but obtained very similar results. We leave a more thorough investigation of dynamic orders as future work.

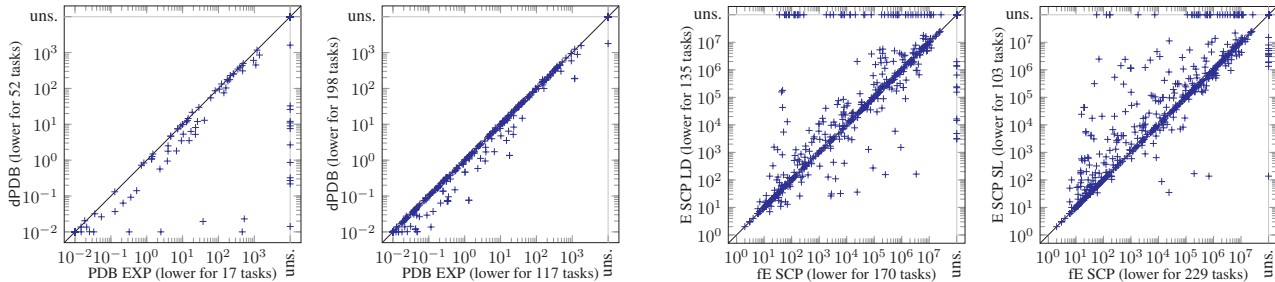

(a) Search time of decoupled search with a single PDB heuristic, using explicit decoupled heuristic (EXP) vs. decoupled PDB (dPDB) computation with factorings F (left) and MM (right).

(b) Expansions of explicit search with the SCP heuristic and factoring MM, comparing using unrestricted PDBs (fE) vs. PDBs restricted to be leaf-disjoint (LD; left) or to affect a single leaf (SL; right).

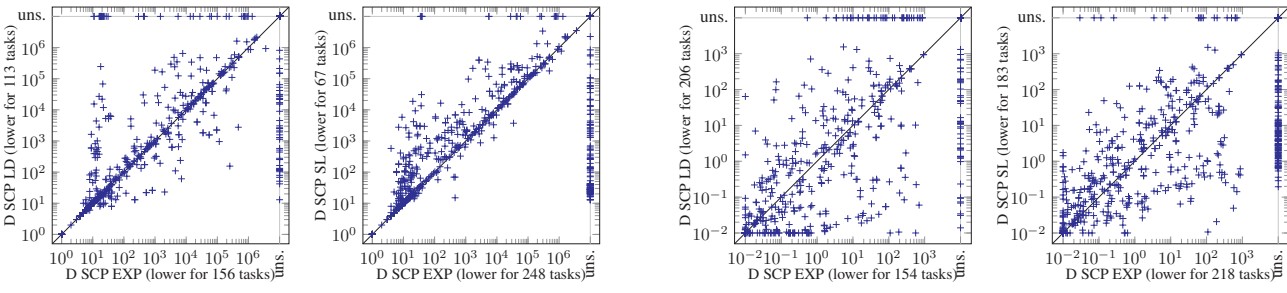

(c) Expansions (2 plots on the left) and search time (2 plots on the right) of decoupled search with the SCP heuristic and factoring MM, comparing explicit decoupled heuristic computation (EXP) vs. leaf-disjoint (LD; left) and single-leaf (SL; right) approximations.

Figure 2: Scatter plots of expansions and search time for different search types, heuristics and factorings.

similar to the results for explicit search which showed that using SL means a severe information loss for CPDBs. With LD, however, coverage increases with all factorings. As observed for the explicit search counter part, the expansions are mostly the same, so LD benefits from an accelerated heuristic computation while computing the same information. For SCP, we again have a different picture, in line with the results for explicit search: LD has a substantially negative effect on coverage and SL can even have a significant positive impact (F and MM). Comparing expansions in the two left-most plots of Figure 2c, we observe that while heuristic quality decreases in more cases than it increases, the reduced search time more often than not compensates this, as evidenced in the two right-most plots of the same figure.

Finally, we compare our results with decoupled PDB-based heuristics to their explicit counterparts and decoupled LM-cut, the state-of-the-art in optimal decoupled search. While the decoupled variants in general cannot compete with their explicit counterparts on the IF and MM factorings, the CPDB-LD and SCP-SL approximations with F solve more tasks than all explicit variants. Decoupled LM-cut solves 299 (F), 198 (IF) and 700 (MM) tasks and is thus better than all CPDB variants; however, SCP-SL solves more tasks with F (304) and MM (709).

## Conclusions

Abstraction heuristics are the state of the art in optimal planning as heuristic search. For decoupled search, the only previous way of using heuristics was via a task compilation depending on the decoupled state, which made using abstraction heuristics practically infeasible. Our paper introduces the explicit decoupled heuristic, which enumerates the member states of the decoupled state, as an alternative way of using arbitrary heuristics for decoupled search. Furthermore, we show how to efficiently compute pattern database heuristics for decoupled states. We show that admissibly combining arbitrary additive pattern collections for a decoupled state without losing information is **NP**-complete. While our algorithm that exactly solves this problem is competitive empirically, we propose two efficient approximations based on imposing restrictions on the pattern collections, which are preferable in most settings, and outperform decoupled search with the LM-cut heuristic in some settings.

In future work, we also want to consider other types of abstractions besides projections. We think that many of our results hold for general partial abstractions, which can also be obtained using the merge-and-shrink framework, for example. Furthermore, the two approximative cases of the explicit decoupled heuristic we presented in this work can be generalized to settings where patterns (or abstractions) are restricted not to a single, but several leaves. These cases will require a more general handling, e.g., using cost-partitioning techniques that also take the pricing function of decoupled states into account.

## Acknowledgements

Silvan Sievers has received funding for this work from the European Research Council (ERC) under the European

Union's Horizon 2020 research and innovation programme (grant agreement no. 817639). Moreover, this work was partially supported by the Wallenberg AI, Autonomous Systems and Software Program (WASP) funded by the Knut and Alice Wallenberg Foundation. Finally, this work was also partially supported by TAILOR, a project funded by the EU Horizon 2020 research and innovation programme under grant agreement no. 952215.

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
