# OpenReview forum: "Additive Pattern Databases for Decoupled Search"
_icaps-conference.org/ICAPS/2022/Workshop/HSDIP — HSDIP 2022_

### Official Review · Reviewer_nyMH · 2022-04-25
**Technical paper with great contribution to decoupled search heuristics**

**Confidence:** 4
**Overall Score:** Accept

**Review:**

The paper introduces a practical means for using PDBs in a decoupled search setting. While previously this was technically possible using a compilation to regular search, this yields approach yields poor results due to the cost-relevant information embedded within the decoupled search space leaves.

The paper is clearly relevant to the HSDIP workshop, polished, and worth accepting. However, it comes across as though the intended audience is limited to those very much "in the know". At times, it was hard to build reasonable intuitions as to why the theoretical results were accomplishing and how the proofs were accomplishing it. This is similarly true for Algorithm 1 (more emphasis in describing what is happening with it, including an example, would help).

I felt similarly with the evaluation, but feel as though this could be remedied with a quick glossary/list of the acronyms used throughout the section.

One question I had after reading the paper is if there are any general lessons learned here for bringing other heuristics to decoupled search. E.g., is there anything generic in the approximation of the NP-Complete approach that could be considered for a heuristic-agnostic approach? Similarly, is there any hope to improve on the compilation due to Gnad and Hoffmann (2018) that preserves more of the decoupled information? (and thus benefiting all heuristics and not just PDB's)

---

### Official Review · Reviewer_E7vB · 2022-04-26
**Solid paper introducing PDB heuristics to Decoupled search**

**Confidence:** 3
**Overall Score:** Accept

**Review:**

This paper provides a detailed study on how to define additive pattern databases for optimal planning using decoupled search. Previous work defining heuristics for decoupled search used the buy-leaves compilation. This paper formalise additive pattern database heuristics and their properties over the structural decomposition imposed by decoupled search, and introduces several approximations to compute additive pattern databases heuristics, which are shown to pay-off experimentally.

It is a solid paper that covers a wide range of ideas in optimal planning. Space limit is an issue, but I remember great illustrative examples  (logistics?) in the first papers introducing decoupled search. I'm curious to know if an illustrative example can show the difference between the different heuristics proposed. Is there a simple PDB whose benefit over buy-leaves heuristics can be easy to follow by a reader?

In terms of Table 1, I'd suggest expanding the caption to define the headings to make it an easier read. I found myself jumping to the text too often to understand the columns and rows.

Overall, this paper is a great fit for the workshop.

---

### Author Response · Authors · 2022-05-20
**Thank your for your comments! We uploaded the version which we intend to submit to SoCS May 23rd.**

Dear reviewers,

thank you for your reviews and please excuse us for not engaging in any discussion!

We believe to have addressed the concerns regarding accessibility of the paper (raised by both of you) by adding an example for decoupled search, also used later for the approximations, although we didn't make it a true running example (e.g., for show-casing how to compute a PDB with Algorithm 1 or how to compute the buy-leaves compilation, which we found less central to the paper).

Regarding the question/commented of reviewer #2: you are certainly right that our results for general decoupled heuristics and we indeed hope that we can use these insights also for other, non-abstraction heuristics, but haven't looked into anything concrete yet.

Regards